# Dysprosium Removal from Water Using Active Carbons Obtained from Spent Coffee Ground

**DOI:** 10.3390/nano9101372

**Published:** 2019-09-25

**Authors:** Lorena Alcaraz, María Esther Escudero, Francisco José Alguacil, Irene Llorente, Ana Urbieta, Paloma Fernández, Félix Antonio López

**Affiliations:** 1National Center for Metallurgical Research (CENIM), Spanish National Research Council (CSIC), Avda. Gregorio del Amo, 8, 28040 Madrid, Spain; alcaraz@cenim.csic.es (L.A.); mevaquero@cenim.csic.es (M.E.E.); falgua@cenim.csic.es (F.J.A.); irene@cenim.csic.es (I.L.); 2Department of Materials Physics, Faculty of Physics, Complutense University of Madrid, s/n, 28040 Madrid, Spain; anaur@fis.ucm.es (A.U.); arana@ucm.es (P.F.)

**Keywords:** dysprosium, activated carbon, spent coffee ground, adsorption

## Abstract

This paper describes the physicochemical study of the adsorption of dysprosium (Dy^3+^) in aqueous solution onto two types of activated carbons synthesized from spent coffee ground. Potassium hydroxide (KOH)-activated carbon is a microporous material with a specific Brunauer–Emmett–Teller (BET) surface area of 2330 m^2^·g^−1^ and pores with a diameter of 3.2 nm. Carbon activated with water vapor and N_2_ is a solid mesoporous, with pores of 5.7 nm in diameter and a specific surface of 982 m^2^·g^−1^. A significant dependence of the adsorption capacity on the solution pH was found, but it does not significantly depend on the dysprosium concentration nor on the temperature. A maximum adsorption capacity of 31.26 mg·g^−1^ and 33.52 mg·g^−1^ for the chemically and physically activated carbons, respectively, were found. In both cases, the results obtained from adsorption isotherms and kinetic study were better a fit to the Langmuir model and pseudo-second-order kinetics. In addition, thermodynamic results indicate that dysprosium adsorption onto both activated carbons is an exothermic, spontaneous, and favorable process.

## 1. Introduction

Nowadays, rare earths (REE) are becoming increasingly interesting due to their essential role in several applications such as permanent magnets, phosphor lamps, rechargeable NiMH batteries, and catalysts, among others [1]. The increasing popularity of hybrid and electric cars, wind turbines, and compact fluorescent lamps is causing an increase in the demand and price of REEs since several compounds of REE, such as neodymium and dysprosium [2], are in smart-batteries that power every electric and hybrid-electric vehicle [3]. In 2010, the European Commission in the report Critical Raw Materials for the European Union considered REEs as the most critical raw materials group. Among them, neodymium (Nd), europium (Eu), terbium (Tb), dysprosium (Dy), and yttrium (Y) were considered the five most critical REEs [1]. It is expected that the demand for Nd and Dy will rise by 700% and 2600% over the next 25 years, respectively [4].

On the other hand, technological development has increased the dumping of electronic waste, which is facilitating the release of significant quantities of these elements along with several other toxic elements into subsoils and groundwater [3]. Hence, the investigation on different adsorption processes is fully pertinent. To the best of our knowledge, maximum acceptable limits for REE in drinking water are not available from any international health organization, nor is sufficient data about their toxicity to human health. However, contamination of the environment by different kinds of toxic species is one of the most serious problems today. Thus, the REE group represents important elements found in the environment and need to be studied at greater depth to understand their effects on human health, and the removal or recovery options from the environment should be deeply investigated [3].

Among the different methods to eliminate contaminants, adsorption onto different carbonaceous materials is an efficient option because of its simplicity and cost-effectiveness [5,6]. Nowadays, the most commonly adopted adsorbent to remove pollutants from water is activated carbon, which is characterized by its high porosity, great surface area, and high degree of surface reactivity [7]. In this sense, the obtention of activated carbons from renewable and cheaper precursors is catching increasing attention from the researchers [7]. Several investigations have been developed about the use of biomass as a precursor, such as residues derived from tea [8,9], coffee [10,11], winemaking waste [12,13], and olives bones [14,15], in order to obtain activated carbon.

In the present work, we obtained activated carbons from spent coffee waste. These activated carbons were both physically and chemically activated. Characterization of both activated carbons and their capacity of dysprosium adsorption were assessed.

## 2. Materials and Methods

### 2.1. Synthesis of the Activated Carbons

Coffee waste recovered from the beverage preparation (spent coffee ground, SCG) were provided by the commercial canteen. The coffee ground used was a mixture 10 (wt%) roasted and 90 (wt%) natural.

SCG were subjected to a hydroalcoholic process in an extraction medium (EtOH:H_2_O, *v*/*v*) 50:50 at 393 K for 30 min in order to obtain the corresponding precursor. Finally, the precursor obtained was turned into activated carbon by both chemical and physical processes.

#### Chemical Activation

To perform the chemical activation, 1 g of the precursor was mixed with 1.5 g of potassium hydroxide (KOH). The homogenized mixtures were placed in alumina crucibles and treated in a Carbolite STF 15 tubular oven (Carbolite Gero, Neuhausen, Germany) at 1123 K for 30 min under a nitrogen carrier (150 mL∙min^−1^). After cooling down to room temperature, the black solid obtained was washed several times with Milli-Q water until neutral pH was achieved. The obtained sample was named AC-CA.

#### Physical Activation

The physical activation of the precursor was carried out in a quartz reactor at 1073 K for 120 min. Approximately 20 g of precursor was added into the reactor. N_2_ with a flow rate of 0.5 mL∙min^−1^ was pumped into the reactor throughout the activation to act as gas carrier during the heating ramp. When the treatment temperature was achieved, deionized water was introduced in the reactor by means of a peristaltic pump with a similar flow rate, hence changing from N_2_ to H_2_O. The obtained sample was named AC-PA.

### 2.2. Characterization of the Activated Carbons

The porous structure of the activated carbons was characterized by N_2_ adsorption at 77 K using an Accelerated Surface Area and Porosimetry System (Micromeritics Instruments Corp, Norcross, GA, USA). The samples were partially degassed at 623 K for 16 h. The specific surface was determined by analyzing the adsorption isotherm via the BET (Brunauer–Emmett–Teller) equation and density functional theory (DFT) models, employing Micromeritics and Quantachrome software (ASAP2020 V03.04, Norcross, GA, USA).

The surface of the activated carbons was examined by field emission scanning electron microscope (FE-SEM,) using a Hitachi S 4800 J microscope (Tokyo, Japan).

Raman experiments and µ-Photoluminescence (µ-PL) were performed at room temperature in a Horiba Jovin Yvon LabRAM HR800 confocal microscope (Horiba, Kyoto, Japan). A He-Cd laser (λ_ex_ = 325 nm) was used as an excitation source for PL, while Raman spectra were recorded under excitation at 632.8 nm line of a He-Ne laser. A charge-coupled device (CCD) detector (Tokyo, Japan) was used to collect the scattered light dispersed by a 600 lines·mm^−1^ grating. The spectral resolution of the system used was 1.5 cm^−1^ for micro-Raman measurements and 0.1 nm for PL measurements.

A VG Microtech model MT500 spectrophotometer (Fison Instruments, Glasgow, UK) with a non-monochromatic MgK_α_1.2 X-Ray source operating at 300W was used. The X-ray photoelectron spectroscopy (XPS) analysis chamber pressure was maintained below 10^−8^ torr during data acquisition. Calibration of the energy scale was performed with an Ag 3d^5/2^ (368.3 eV) standard. The peaks were fitted using a Gaussian-Lorentzian mixed function after a Shirley background subtraction. C1s sp^2^ at 284.5 eV was used as binding energy (BE) reference.

### 2.3. Adsorption Experiments

Dysprosium adsorption by both activated carbons (AC-CA and AC-PA) was carried out via batch experiments. Stock solution of dysprosium (1000 mg·L^−1^) were prepared by dissolving Dy(NO_3_)_3_·xH_2_O (Merck KGaA, Damstadt, Germany) in MilliQ water. Then, stock solution was diluted to obtain the different concentrations. The temperature was controlled using a Selecta Termotronic thermostat bath equipped with multiple Lab Companion MS-52M stirrers. Aliquots (1 mL) of the solution were extracted every 15 min and filtered through a syringe filter with a 0.22 µm pore and 13 mm diameter. The pH of the solutions was adjusted with HCl (0.1 M) until the desirable value was achieved. Dysprosium content in the solution was analyzed by Inductively Coupled Plasma-Optical Emission Spectrometry (ICP-OES) using an Agilent ICP-OES (Agilent Technologies, Sana Clara, CA, USA), model 5100 VDV (Vertical Dual View).

The dysprosium adsorbed percentage and the adsorption capacity at each time t (q_t_ (mg·g^−1^)) were calculated by the following equations:(1)$$\mathrm{Adsorbed}\;\mathrm{Dy}=\frac{\left(c_{0}-c_{e}\right)}{c_{0}}\cdot 100$$
(2)$$q_{t}=\frac{\left(C_{0}-C_{e}\right)\;\cdot \;V}{m}$$

The equilibrium adsorption isotherm data were plotted using the linear forms of Langmuir (Equation (3)), Freundlich (Equation (4)), and Temkin (Equation (5)) models [16]:

Langmuir:(3)$$\frac{C_{e}}{q_{e}}=\frac{1}{q_{m}\cdot b}+\frac{1}{q_{m}}\cdot c_{e}$$

Freundlich:(4)$${\ln\;q}_{e}={\ln\;K}_{F}+\frac{1}{n}\cdot {\ln\;C}_{e}$$

Temkin:(5)$$q_{e}=B\cdot \ln A_{T}+B\cdot \ln c_{e}$$

Kinetics experiments were analyzed using the pseudo-first-order (Equation (6)) and pseudo-second-order (Equation (7)) models, and were carried out at different temperatures:

Pseudo-first order [17]:(6)$$\ln\left(q_{e}-q_{t}\right)=\ln q_{e}-K_{1}\cdot t$$

Pseudo-second order [18]:(7)$$\frac{t}{q_{t}}=\frac{1}{K_{2}\cdot q_{e}^{2}}+\frac{1}{q_{e}}\cdot t$$

Activation energy (E_a_) can be calculated from the Arrhenius equation (Equation (8)) expressed as:(8)k2=A·e(−EaR·T)

Enthalpy change (ΔH^0^) and entropy change (ΔS^0^) were calculated from the slope and intercept of a plot of log (q_e_/c_e_) versus 1/T [19] using the equations (Equation (9)) and (Equation (10)):(9)$$\log\frac{q_{e}}{c_{e}}=\frac{\Delta S^{0}}{2.303R}+\frac{\Delta H^{0}}{2.303\mathrm{RT}}$$
(10)$$\Delta G^{0}=\Delta H^{0}-T\Delta S^{0}$$

## 3. Results and Discussion

### 3.1. Characterization of the Activated Carbons

#### 3.1.1. Textural Properties

The pore volume, pore size, and the total specific surface area on AC-CA and AC-PA activated carbons were determined by the BET equation. AC-CA with a basically microporous structure were obtained, as can be seen in Table 1, with the total pore volume (V_p_) very similar to the volume of micropores (W_0_) a pore diameter (D_p_) of <3.29 nm (micropores). The BET surface area is 2330 m^2^·g^−1^. In the AC-PA, the volume of micropores is much lower than the total pore volume, which indicate that it is a mesoporous material, with pores of 4.8 mm. The non-microporous surface is much larger that the microporous surface, in accordance with the previous data. The specific BET surface is 981 m^2^·g^−1^.

Texturally, both activated carbons are very different. Consequently, they should exhibit different behavior in the Dy adsorption process.

The N_2_ adsorption isotherm of AC-CA (Figure 1a) is of type I, according to the International Union of Pure and Applied Chemistry (IUPAC) classification [20]. This isotherm is characteristic of microporous solids. Figure 1b shows the pore size distribution (Barrett-Joyner-Halenda (BJH) desorption). The average pore diameter is 3.29 nm. In the case of AC-PA, the adsorption isotherm (Figure 1c) is of type IV, with characteristics of solid that present aggregates of particles in the form of slit-shaped. The pore size distribution (Figure 1d) shows an average pore diameter of 5.68 nm.

#### 3.1.2. Scanning Electron Microscopy (SEM)

The surface characteristics of the activated carbons were analyzed using the SEM technique. Different morphologies can be clearly observed. The chemically activated carbons (Figure 2a) show homogeneous surface morphology with a large number of pores caused by the chemical interaction between KOH and the precursor surface. On the other hand, AC-PA (Figure 2b) exhibit a surface less porous than in the case described above, with a pitted and cracked surface typical of the physically activated carbons [21].

#### 3.1.3. Raman Spectroscopy

Characterization of both active carbons was carried out by Raman spectroscopy. Raman spectroscopy is a useful technique for carbon materials, because the spectral shape drastically changes not only due to the kind of abundant allotropic forms of carbon, but also to the fine structural changes of the individual allotrope [22]. In this sense, polycrystalline graphites exhibit two sharp peaks: G-band (at around 1580 cm^−1^) and D-band (at around 1355 cm^−1^). These bands are generally attributed to E_2g_ and A_1g_ in-plane vibration modes, respectively, and the graphitization degree of a carbon material is generally characterized by the I_G_/I_D_ value in the Raman spectra. Both vibration modes are represented schematically in Figure 3. In part A of the scheme, the breathing mode A_1g_ (D-band) is shown, while part B illustrates the vibration mode E_2g_ (G band) [23,24]. On the other hand, in amorphous carbon, a broad band around 1550 cm^−1^ overlapped with a broader band around 1400 cm^−1^ is observed [22,25].

Comparative normalized spectra of AC-CA and AC-PA activated carbons are shown in Figure 4. Both samples exhibit two Raman bands peaked at 1595 cm^−1^ and 1334 cm^−1^. A broadening of the Raman bands can be appreciated in the AC-CA sample in comparison to the AC-PA sample. In addition, the Raman bands centered at 1334 cm^−1^ exhibit the higher intensity in both cases. This effect is more noticeable in the AC-PA activated carbon spectrum.

In the present case, the I_G_/I_D_ (AC-PA) > I_G_/I_D_ (AC-CA), which indicate the highest graphitization degree of the AC-PA activated carbon sample. Previous investigations carried out with carbon nanomaterials reported that the adsorption capacity is related with the graphitization degree [24].

### 3.2. Adsorption Experiments

We have studied the influence of three different parameters on the adsorption capacity of both types of activated carbons: Solution pH, Dy concentration, and activated carbon amount.

#### 3.2.1. Influence of the Solution pH

In order to analyze the influence of the solution pH on the adsorption process, experiments were carried out at three different pH values (3 to 5). For this purpose, 30 mg of the activated carbons were added to 200 mL solution containing 5 mg·L^−1^ of Dy ions. The adsorbed dysprosium amounts versus the contact time at different pH values for both types of AC are plotted in Figure 5.

In both cases (AC-CA and AC-PA), the adsorption capacity increases when the pH increases from 3 to 4 and then decreases. As previously reported [26], dysprosium may be present in the solution either as an isolated cation (Dy^3+^) or as hydroxide cation (Dy(OH)^2+^). Consequently, the behavior in aqueous solution is a complex phenomenon. According to Qaader et al. [26], at a pH between 1 and 4, the predominant species is Dy^3+^, while at higher pH values Dy^3+^ ions start to become hydrolyzed, leading to the formation of other species such as Dy(OH)^2+^.

These species are weakly adsorbed as compared to Dy^3+^ ions. Therefore, the adsorption of dysprosium starts to decrease above pH 5, and the highest adsorption was found at value of pH of 4, obtaining a maximum adsorption percentage of 94% for AC-CA and practically 100% for AC-PA. The adsorption capacity of AC-PA was higher than AC-CA for all pH values investigated.

#### 3.2.2. Influence of the Dysprosium Concentration

The influence of dysprosium concentration has been studied with three solutions with Dy concentrations of 2.5, 5, and 10 mg/L, and a fixed amount (30 mg) of activated carbon. All the experiments were carried out at room temperature. The results are shown in Figure 6.

In both cases (AC-CA and AC-PA), the adsorbed Dy percentage decreases with increasing RE concentration. In the case of AC-PA, the adsorption percentages are quite high, ranging from 99% for the lowest Dy concentration to 80% for the largest. However, the AC-CA samples show lower adsorption percentages. In fact, a significant adsorption, 96%, is obtained (similar to that of AC-PA) only for the solution with the lowest Dy concentration. For the solutions with 5 and 10 mg·L^−1^ of Dy, much lower adsorption capacities are observed, between 20 and 30%.

#### 3.2.3. Influence of the Activated Carbon Amount

Finally, the influence of the adsorbent amount was studied. Solutions of Dy with 5 mg·L^−1^ of concentration were put in contact with different amounts of both ACs (5, 15, 30, and 60 mg). The results are plotted in Figure 7.

As for the other parameters, the best results are obtained for the AC-PA. As expected, the larger the amount of adsorbent, the larger the adsorption percentage. However, the behavior of the two kinds of samples is significantly different. In the case of the AC-PA, the adsorption percentage increases drastically (from 57% to 99%) when the AC amount increases from 5 mg to 15 mg. Then, it remains practically constant with the amount of adsorbent. However, in the case of the AC-CA, the adsorption depends strongly on the adsorbent amount, going from 10% (5 mg of the AC) to 96% (60 mg of the AC). Only the largest adsorbent amount the adsorption percentage is comparable to those of AC-PA, and even in this case, the time required to reach a similar adsorption percentage is much longer. Comparable values are reached only for the largest adsorbent amount (60 mg) and times around 120 min.

#### 3.2.4. Equilibrium Isotherms

As observed from the previously shown adsorption results (Figure 4, Figure 5 and Figure 6), the equilibrium adsorption percentage is reached after a period of time ranging from approximately 50 to 120 min depending on the experiment conditions. For the sake of the kinetic study, we will adopt those obtained at 120 min as equilibrium values since, at this time, the equilibrium has been achieved in all the experimental conditions.

Equilibrium isotherms were measured modifying the adsorbent amount and using Equations (3)–(5). In Table 2, the calculated parameters and the corresponding correlation coefficients are shown. For both ACs, the best correlation is obtained with the Langmuir model. According to this model, the maximum adsorption capacity values (q_m_) calculated were 28.11 mg·g^−1^ and 29.05 mg·g^−1^ for AC-CA and AC-PA, respectively. These results are similar to those obtained experimentally (31.26 mg·g^−1^ and 33.52 mg·g^−1^). As expected, according to the previously discussed results, the AC-PA showed the greatest adsorption capacity. The values calculated for Langmuir nondimensional factors (RL) are 0.02 and 0.03 for AC-CA and AC-PA respectively, which indicates that the dysprosium adsorption is a favorable process.

#### 3.2.5. Effect of the Temperature, Kinetic and Thermodynamic Study

Experiments at different three temperatures (303, 318, and 333 K) were carried out to evaluate the influence of temperature on adsorption process. For this experiment, 30 mg of the AC were put in contact with a Dy dissolution with a concentration of 5 mg·L^−1^. As can be seen in Figure 8, the behavior of both types of samples is considerably different. In the AC-PA samples, temperature does not seem to play a significant role (within the temperature interval under consideration). For all temperatures investigated, the equilibrium value is reached after a relatively short time (around 25 min). In the case of the AC-CA sample, at the lowest temperature, a very low adsorption degree is reached (27%). At 318 K, the adsorption percentage is closer to that obtained for AC-PA samples. At 333 K, similar values are obtained for both types of activated carbons.

Adsorption kinetics studies were carried out using Equations (6) and (7). The calculated parameters from the corresponding fits are included in Table 3. For both type of samples, the best fitted results were found for a pseudo-second-order model. With respect to the kinetic reaction constants, k_2_ increases as the temperature increases, which indicates that the temperature favors the adsorption process. In fact, if we consider the k_2_ values obtained for AC-CA samples, the influence of temperature seems to be larger than for the AC-PA samples, in agreement with the results described from Figure 8.

As explained in Section 2.2, activation energy was estimated the linear form Arrhenius equation plot ln k_2,obs_ versus 1/T. The value of the activation energy will give us information on the character of the adsorption process. For physical adsorption process, the reactions are reversible, the equilibrium are rapidly achieved, and the activation energies are correspondingly low in the range from 5 to 40 kJ·mol^−1^. On the contrary, chemical adsorption involves stronger forces and hence requires higher activation energies (40 to 800 kJ·mol^−1^) [27]. In our case, the calculated activation energies were 10.90 kJ·mol^−1^ and 7.83 kJ·mol^−1^ for AC-CA and AC-PA, respectively, suggesting a physisorption process.

Table 4 summarizes the thermodynamic parameters calculated from Equations (8)–(10). The negative values obtained for the enthalpy change indicate that the dysprosium adsorption is an exothermic process in all the samples studied. Moreover, the adsorption process is spontaneous and favorable at the different temperatures studied for both ACs, as indicated by the free energy change [28]. Finally, the value calculated for the entropy change was positive in both cases, indicating that the entropy of the system increased during the adsorption, i.e., as mentioned previously, it is a spontaneous process. These results described are indicative of the affinity of the ACs toward Dy ions [27]. Besides, the positive values suggest an increase in adsorbate concentration in the solid-liquid interface (an increase in adsorbate concentration onto the solid phase) [29].

### 3.3. X-Ray Photoelectron Spectroscopy (XPS)

The shape of high-resolution Dy 4d XPS spectra and its binding energy were carefully analyzed (Figure 9). The Dy spectra exhibit doublet components due to the electrostatic interactions of the 4d hole and 4f electrons with spin-orbit splitting of 4d^5/2^ and 4d^3/2^ states. In the case of Dy(NO_3_)_3_, the characteristic doublet peak appears at 154.2 eV for 4d^5/2^ and 157.4 eV for 4d^3/2^, which is according to the standard oxidation state of +3 [30]. The binding energy of Dy 4d^5/2^ for samples AC-PA:Dy and AC-CA:Dy are 155.0 eV and 153.9 eV, respectively. These results confirm that the Dy remains in the state of Dy^3+^ after treatment, but in the case of the sample AC-CA:Dy, its BE value is very close to the obtained for Dy(NO_3_)_3_ standard, which means that there is no significant difference in the chemistry of Dy in both samples. However, in the case of AC-PA:Dy, there is a significant shift to higher binding energies. This could suggest a higher interaction between Dy and C after this treatment. The obtained results can be indicative of a chemisorption process in the case of AC-PA activated carbon, despite the results obtained from thermodynamic studies. Nevertheless, in the case of the AC-CA, the adsorption could be a physisorption process.

### 3.4. Photoluminescence Spectroscopy (PL)

Like most RE, dysprosium cations have a strong visible luminescent emission related to different intra-ionic transitions [31]. Luminescence spectra of Dy^3+^ active ions are composed of three characteristic bands centered at 480, 575, and 664 nm. These emissions can be attributed to ^4^F_9/2_-^6^H_15/2_, ^4^F_9/2_-^6^H_13/2_, and ^4^F_9/2_-^6^H_11/2_ transitions, respectively [32]. Among them, the two first are the dominant components in the spectra [33]. The emission intensity of Dy^3+^ ions may strongly depend on the host matrix, which may significantly influence the radiative and non-radiative properties (multiphonon relaxation as well as energy transfer) leading to different luminescent behavior [32]. Among the different emissions mentioned above, the ^4^F_9/2_-^6^H_13/2_ transition is the most sensitive and the ^4^F_9/2_-^6^H_15/2_ transition is the less sensitive to the matrix [34].

The spectrum shown in Figure 10 corresponds to the AC-PA:Dy samples. The spectrum exhibits a dominant band peaked at 572 nm and a less intense band centered at around 478 nm. These bands can be attributed to ^4^F_9/2_-^6^H_13/2_ and ^4^F_9/2_-^6^H_15/2_ transitions, respectively. No PL emission could be detected from the AC-CA:Dy samples. In the AC-PA:Dy samples, the dominant band is the ^4^F_9/2_-^6^H_13/2_, sometimes referred as hypersensitive transition due to the high sensitivity to the host. Therefore, structural differences between both types of carbons could be behind the quenching of the luminescence observed in the AC-CA samples.

## 4. Conclusions

Dysprosium adsorption experiments exhibit that the variation of the solution pH realizes a great influence in the process, obtaining the maximum adsorption at pH 4. As an expected, a decreases of the solution Dy concentration increases the adsorption percentage. In addition, in the case of chemically activated carbon, the Dy absorbed amount gradually increases with the adsorbent dosage. This effect is not observed in the case of physically activated carbon, where with the exception of the lowest amount used, the adsorption percentage remains practically constant. Finally, the variation of the temperature slightly improves the adsorption process. Despite this, physically activated carbon exhibits a great adsorption capacity in all cases. The adsorption isotherms and the kinetic study were better fit with the Langmuir model and pseudo-second-order kinetics, respectively, for both investigated activated carbons. So, dysprosium adsorption onto both activated carbons is an exothermic, spontaneous, and favorable process, as indicated the obtained thermodynamic results. XPS measurements indicate that Dy ions remain in the +3 oxidation state after the adsorption process. Moreover, in the case of the AC-PA:Dy sample, a significant shift to higher binding energies was found indicating higher interaction between Dy ions and the AC. The PL spectrum of the Dy loaded AC-PA sample exhibit an emission intensity centered at 572 nm attributed to the Dy^3+^ ions. So, in the case of the AC-PA activated carbon, the Dy adsorption could be a chemisorption process.

## Figures and Tables

**Figure 1 nanomaterials-09-01372-f001:**
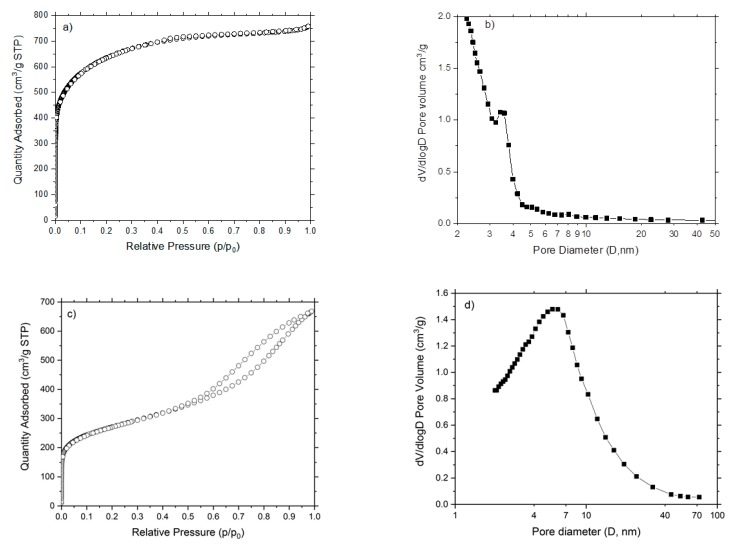
Adsorption isotherms and pore size distributions of (**a**,**b**) AC-CA and (**c**,**d**) AC-PA.

**Figure 2 nanomaterials-09-01372-f002:**
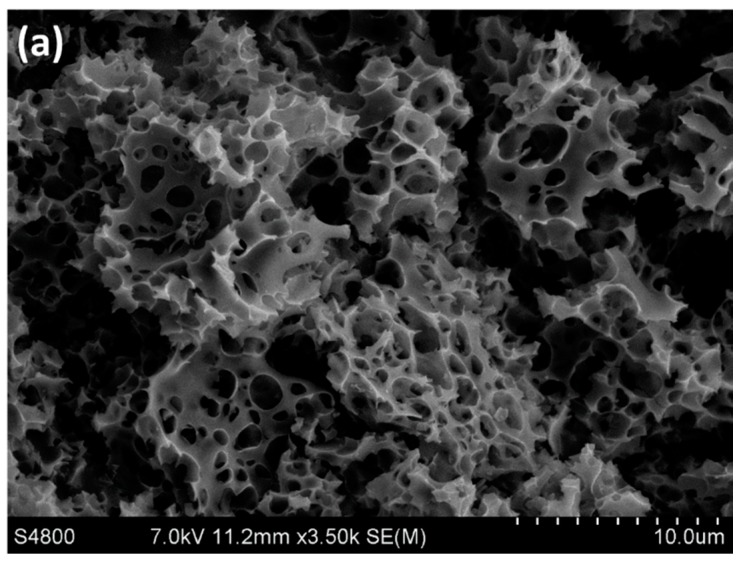
SEM micrographs of (**a**) AC-CA and (**b**) AC-PA samples.

**Figure 3 nanomaterials-09-01372-f003:**
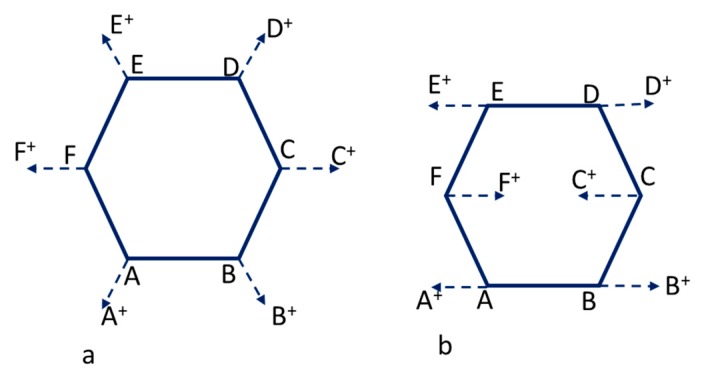
Scheme of the vibration modes: (**a**) Breathing mode A_1g_ (D-peak) and (**b**) Vibration mode E_2g_ (G-peak).

**Figure 4 nanomaterials-09-01372-f004:**
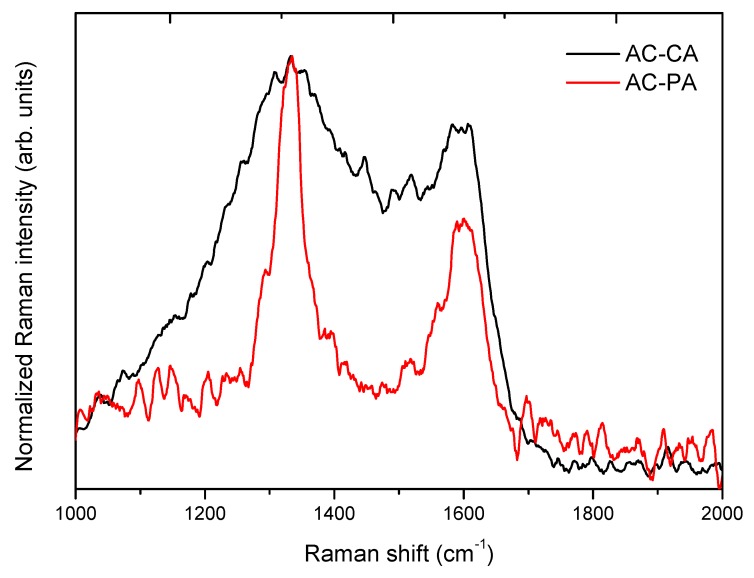
Normalized Raman spectra of both activated carbons.

**Figure 5 nanomaterials-09-01372-f005:**
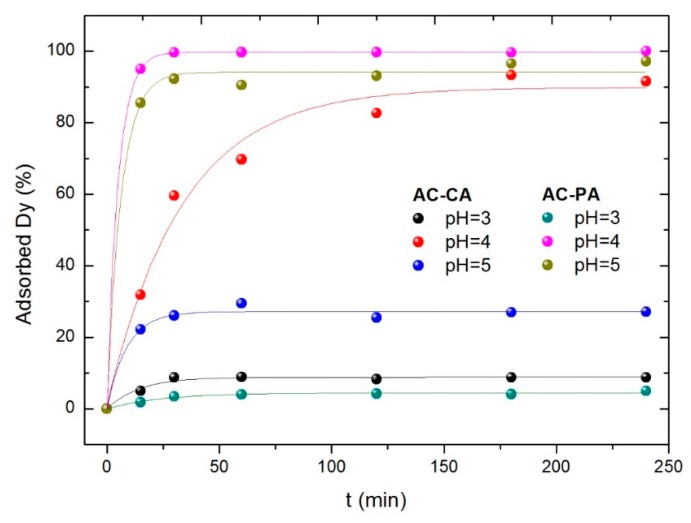
Effect of the solution pH for AC-CA and AC-PA.

**Figure 6 nanomaterials-09-01372-f006:**
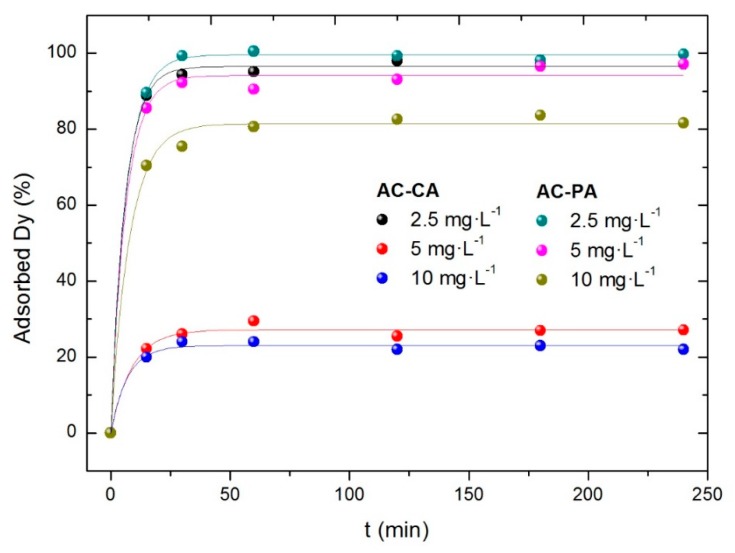
Effect of the dysprosium concentration in the adsorption percentage.

**Figure 7 nanomaterials-09-01372-f007:**
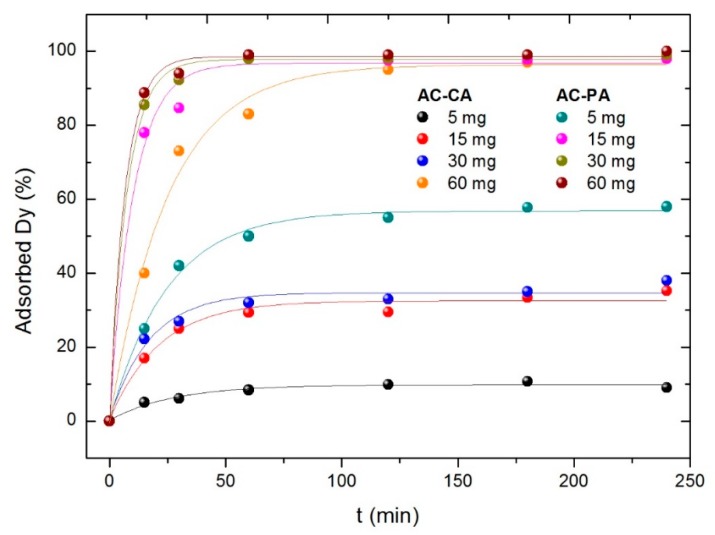
Variation of the adsorption percentage with the adsorbent amount.

**Figure 8 nanomaterials-09-01372-f008:**
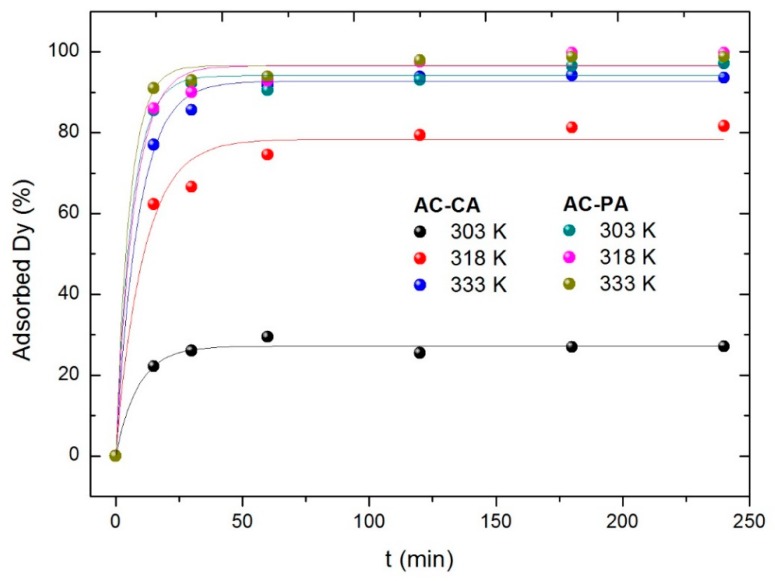
Dysprosium (Dy) adsorption percentage with the temperature variation.

**Figure 9 nanomaterials-09-01372-f009:**
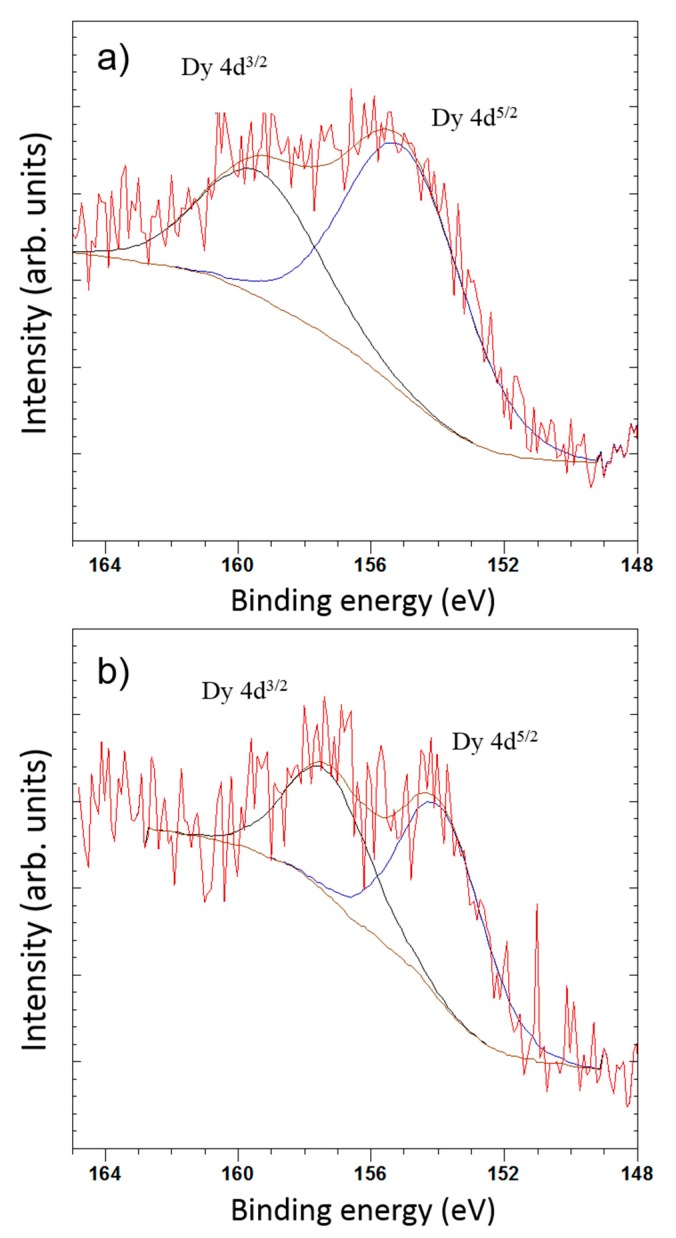
X-ray photoelectron spectroscopy (XPS) deconvolution spectra of Dy 4d peak of (**a**) AC-PA:Dy and (**b**) AC-CA:Dy.

**Figure 10 nanomaterials-09-01372-f010:**
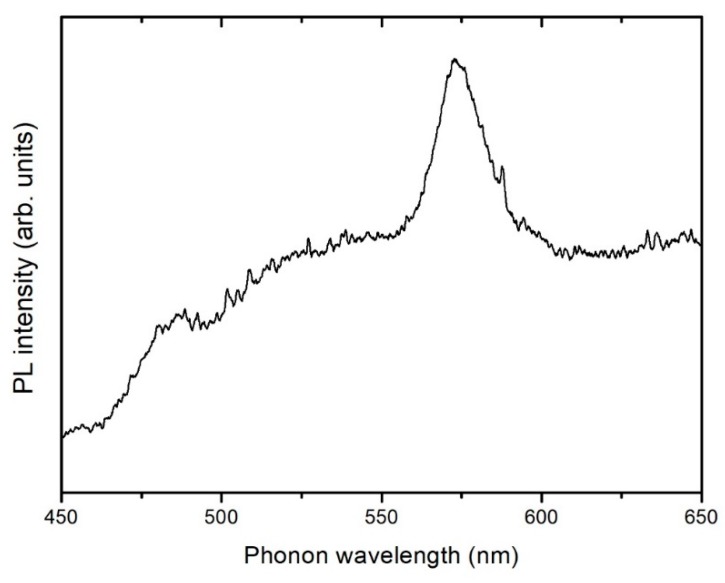
Photoluminescence (PL) spectrum of adsorbed dysprosium onto the AC-PA activated carbon.

**Table 1 nanomaterials-09-01372-t001:** Textural characterization of the activated carbons.

Sample	V_T_ (cm^3^·g^−1^)	V_μp_ (cm^3^·g^−1^)	D_p_ (nm)	S_μS_ (m^2^·g^−1^)	NS_μs_ (m^2^·g^−1^)	S_BET_ (m^2^·g^−1^)
**AC-QA**	1.17	1.07	3.29	2265.0	65.6	2330.6
**AC-PA**	1.03	0.03	5.68	244.8	736.8	981.6

**Table 2 nanomaterials-09-01372-t002:** Calculated isotherm parameters for the Langmuir, Freundlich, and Temkin linear models.

Sample	Langmuir				Freundlich			Temkin		
	q_m_ (mg·g^−1^)	b (L·mg^−1^)	R_L_	R^2^	K_F_ (L·g^−1^)	1/n	R^2^	A_T_	b_T_	R^2^
**AC-CA**	28.11	6.42	0.02	0.996	24.82	0.08	0.713	0.23	0.16	0.582
**AC-PA**	29.05	10.5	0.03	0.998	81.44	0.17	0.722	0.79	0.10	0.800

**Table 3 nanomaterials-09-01372-t003:** Calculated kinetics parameters at different temperatures.

Sample	T (K)	Pseudo-First-Order			Pseudo-Second-Order		
		k_1_	q_e_	R^2^	k_2_ (·10^-3^)	q_e_	R^2^
**AC-CA**	303	0.016	17.693	0.784	0.795	32.362	0.997
	318	0.017	19.470	0.830	1.012	32.467	0.995
	333	0.057	19.931	0.933	10172	33.670	0.999
**AC-PA**	303	0.020	8.194	0.740	0.932	28.490	0.999
	318	0.023	8.366	0.965	1.132	30.120	0.999
	333	0.047	21.270	0.785	1.232	32.362	0.999

**Table 4 nanomaterials-09-01372-t004:** Calculated thermodynamic kinetics parameters for both activated carbons (ACs).

Sample	T (K)	−ΔH^0^ (kJ·mol^−1^)	ΔS^0^ (J·mol^−1^·K^−1^)	−ΔG^0^ (kJ·mol^−1^)
**AC-CA**	303	79.18	327.64	178.45
	318			183.37
	333			188.29
**AC-PA**	303	159.65	628.29	350.02
	318			359.44
	333			368.87

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
