# Peer review of "Dysprosium Removal from Water Using Active Carbons Obtained from Spent Coffee Ground"

_nanomaterials, 2019, doi:10.3390/nano9101372_

Round 1

Reviewer 1 Report

Manuscript ID nanomaterials-578199

The paper describes the study of the dysprosium (Dy3+) adsorption from the aqueous solution using two types of active carbons prepared from spent ground coffee. The issues discussed are very important from the point of view of environmental protection. While reading the manuscript, some drawbacks are noted, which should be taken into account during the revision of the paper:

line 63: the authors should explain the effect of the hydro-alcoholic extraction process.

lines 79-80: please correct the sentence.

What was the rate of temperature rise to 1123 K (chemical activation) or 1073 K (physical activation)? This is important because the rate of temperature increase has a significant impact on porosity creation.

The XPS analysis should be mentioned in 2.2 paragraph.

line 143: the BET is not a technique, but equation, please correct the statement.

line 143-145: “The AC-CA with a basically microporous structure were obtained, as can be appreciated in Table 1, with the total pore volume (Vp) very similar to the volume of micropores (W0) a pores diameter (Dp) of <3.29 nm (micropores).” – the statement is not clear. It requires change.

 Moreover, pores of diameter 3.29 nm (line 145) are (according to the theory ) narrow mesopores but not micropores – adequate references should be added and the statement about micropores should be modified.

line 176: Raman spectroscopy should be mentioned in 2.2 paragraph.

lines 201-202 : such information should be included in the Adsorption experiments section

Table 3: the numbers should contain full stops, not commas – please correct

line 302:L 2.2 section does not include information about the Arrhenius plot. please check the numbering

line 338: Photoluminescence spectroscopy should be mentioned in 2.2 paragraph.

Author Response

Dysprosium removal from water using active carbons obtained from spent coffee ground, by L. Alcaraz, M.E. Escudero, F.J. Alguacil, I. LLorente, A. Urbieta, P. Fernandez, and F.A. López, (Manuscript ID nanomaterials-578199).

Response to referee 1

We would like to thank the referee for his/her comments. In the following lines, we answer his/her questions on a point by point basis and indicate how the text has been modified.

The paper describes the study of the dysprosium (Dy3+) adsorption from the aqueous solution using two types of active carbons prepared from spent ground coffee. The issues discussed are very important from the point of view of environmental protection. While reading the manuscript, some drawbacks are noted, which should be taken into account during the revision of the paper:

line 63: the authors should explain the effect of the hydro-alcoholic extraction process

The aim of the hydro alcoholic extraction process was the polyphenols recover present in coffee wastes which have value added, as indicated in the following reference [1]:

Ramón-Gonçalves, M.; Gómez-Mejía, E.; Rosales-Conrado, N.; León-González, M.E.; Madrid, Y. Extraction, identification and quantification of polyphenols from spent coffee grounds by chromatographic methods and chemometric analyses. Waste Manag. 2019, 96, 15–24.

lines 79-80: please correct the sentence.

There is a mistake. In the revised version this paragraph has been modified:

“… means of a peristaltic pump, with a similar flow rate, hence changing from N2 to H2O. The obtained sample is named as AC-PA.”

What was the rate of temperature rise to 1123 K (chemical activation) or 1073 K (physical activation)? This is important because the rate of temperature increase has a significant impact on porosity creation.

As the referee indicate, the temperature is an important factor which has a significant impact on the sample porosity. In general, an increase of the temperature leads to higher porosity samples. However, the increase of the temperature results high burn-off, obtaining a low yield in the process. In the present work, the authors synthesized activated carbons with a high specific surface and porosity and a significant yield.

The XPS analysis should be mentioned in 2.2 paragraph.

The authors are agreeing with the referee. In the revised version, the technique is mentioned in the 2.2. section.

line 143: the BET is not a technique, but equation, please correct the statement.

The referee is right. In the revised version the mistake has been modified.

Line 144: “… carbons were determined by the BET equation. …”

line 143-145: The AC-CA with a basically microporous structure were obtained, as can be appreciated in Table 1, with the total pore volume (Vp) very similar to the volume of micropores (W0) a pores diameter (Dp) of <3.29 nm (micropores).” – the statement is not clear. It requires change.  Moreover, pores of diameter 3.29 nm (line 145) are (according to the theory) narrow mesopores but not micropores – adequate references should be added and the statement about micropores should be modified.

The referee is right. According to IUPAC nomenclature [2] the obtained results indicate a mesoporous structures. The paragraph had also been modified in the revised version:

“AC-CA exhibit basically mesoporous structure, with a pores diameter (Dp) of 3.29 nm. The BET surface area is 2330 m2·g-1. In the AC-PA the volume of micropores is much lower than the total pore volume, which indicate that it is a mesoporous material, with pores of 4.8 mm.”

Union, I.; Pure, O.F.; Chemistry, A. Recommendations for the characterization of porous solids (Technical Report). Pure Appl. Chem. 1994, 66, 1739–1758.

 line 176:Raman spectroscopy should be mentioned in 2.2 paragraph.

The authors are agreeing with the referee. In the revised version, the technique is mentioned in the 2.2. section.

 Table 3: the numbers should contain full stops, not commas – please correct

In the revised version, commas have been replacing by full stops

line 302: L 2.2 section does not include information about the Arrhenius plot. please check the numbering

As explained in line 130 in the revised version, Arrhenius equation is:

In the linear form, ln k2,obs versus 1/T can be used to obtain the Ea. In the revised versión, the authors modified the subsequent lines:

Lines 312-313: “ As explained in section 2.2, activation energy was estimated the linear form Arrhenius equation plot ln k2,obs versus 1/T.”

line 338: Photoluminescence spectroscopy should be mentioned in 2.2 paragraph.

The authors are agreeing with the referee. In the revised version, the technique is mentioned in the 2.2. section.

Reviewer 2 Report

This manuscript is described that two types of activated carbon as an absorber from spent coffee grounds was prepared through chemical and physical method and studied dysprosium removal behavior from water. Activated carbons were characterized by BET and FE-SEM. Dysprosium removal behavior from water were investigated by batch experimental, μ-Photoluminescence, Raman spectra, and XPS. And their adsorption behavior was calculated by the Langmuir, Freundlich, and Temkin linear model. This paper is coupled with very concise process and method and focusing the points of uncertainty. 

Would you concretely explain why dysprosium adsorption behavior of AC-CA samples is lower than that of AC-PA? And also, why does it show that adsorption rate of AC-CA is increased with temperature? You should be explained these.

Author Response

We would like to thank the referee for his/her comments. In the following lines, we answer his/her questions.

Would you concretely explain why dysprosium adsorption behaviour of AC-CA samples is lower than that of AC-PA?

As indicated in the manuscript, the obtained results from XPS indicated higher interaction between Dy ions and the AC-PA. So, in the case of the AC-PA activated carbon the Dy adsorption could be a chemisorption process.

And also, why does it show that adsorption rate of AC-CA is increased with temperature? You should be explained these.

Despite of the obtained results from thermodynamic studies which indicate that the adsorption process is an exothermic process, in the case of the AC-CA, dysprosium adsorption increases with the temperature. However, in the case of the AC-PA, the increase of the adsorption with the temperate is miserable. This effect is typical of physisorption processes onto activated carbons, as in the present case.